# Impact of H&E Stain Normalization on Deep Learning Models in Cancer Image Classification: Performance, Complexity, and Trade-Offs

**DOI:** 10.3390/cancers15164144

**Published:** 2023-08-17

**Authors:** Nuwan Madusanka, Pramudini Jayalath, Dileepa Fernando, Lasith Yasakethu, Byeong-Il Lee

**Affiliations:** 1Digital Healthcare Research Center, Pukyong National University, Busan 48513, Republic of Korea; nuwanmadusanka@hotmail.com; 2Institute of Biochemistry, Faculty of Mathematics and Natural Science, University of Cologne, 50923 Cologne, Germany; pjayalat@smail.uni-koeln.de; 3School of Computer Science and Engineering, Nanyang Technological University, Singapore 639798, Singapore; dileepa.fernando@ntu.edu.sg; 4Department of Software Engineering, Sri Lanka Technological Campus (SLTC), Padukka 10500, Sri Lanka; lasithy@sltc.ac.lk; 5Division of Smart Healthcare, College of Information Technology and Convergence, Pukyong National University, Busan 48513, Republic of Korea; 6Department of Industry 4.0 Convergence Bionics Engineering, Pukyoung National University, Busan 48513, Republic of Korea

**Keywords:** deep learning, stain normalization, computational complexity, cancer classification, image classification

## Abstract

**Simple Summary:**

This research study investigates the impact of stain normalization on deep learning models for cancer image classification by evaluating model performance, complexity, and trade-offs. The primary objective is to assess the improvement in accuracy, performance, and resource optimization of deep learning models through the standardization of visual appearance in histopathology images using stain normalization techniques, alongside batch size and image size optimization. The findings provide valuable insights for selecting appropriate deep learning models in achieving precise cancer classification, considering the effects of H&E stain normalization and computational resource availability. This study contributes to the existing knowledge on the performance, complexity, and trade-offs associated with applying deep learning models to cancer image classification tasks.

**Abstract:**

Accurate classification of cancer images plays a crucial role in diagnosis and treatment planning. Deep learning (DL) models have shown promise in achieving high accuracy, but their performance can be influenced by variations in Hematoxylin and Eosin (H&E) staining techniques. In this study, we investigate the impact of H&E stain normalization on the performance of DL models in cancer image classification. We evaluate the performance of VGG19, VGG16, ResNet50, MobileNet, Xception, and InceptionV3 on a dataset of H&E-stained cancer images. Our findings reveal that while VGG16 exhibits strong performance, VGG19 and ResNet50 demonstrate limitations in this context. Notably, stain normalization techniques significantly improve the performance of less complex models such as MobileNet and Xception. These models emerge as competitive alternatives with lower computational complexity and resource requirements and high computational efficiency. The results highlight the importance of optimizing less complex models through stain normalization to achieve accurate and reliable cancer image classification. This research holds tremendous potential for advancing the development of computationally efficient cancer classification systems, ultimately benefiting cancer diagnosis and treatment.

## 1. Introduction

Histopathology image analysis plays a crucial role in cancer diagnosis and treatment. With the advancements in deep learning (DL) techniques, the use of DL models for cancer image classification has shown promising results [1]. However, the performance and reliability of these models heavily rely on the quality and consistency of input data. Histology images are commonly stained using Hematoxylin and Eosin (H&E) to enhance tissue contrast and aid in visual interpretation. However, variations in staining protocols and equipment can introduce visual inconsistencies among images, potentially affecting the performance of DL models [2,3].

Histology image stain normalization techniques have emerged as a means to address these visual inconsistencies by standardizing the appearance of images. By applying stain normalization methods, it is possible to remove or reduce staining variations and ensure a consistent visual representation of the underlying tissue structures. This normalization process holds the potential to improve the accuracy, reliability, and resource utilization of DL models for cancer image classification tasks.

In this research study, we investigate the impact of histology image stain normalization on the of DL models performance in cancer image classification [4,5]. We conduct a comprehensive analysis using Generative Adversarial Network (GAN)-based stain normalization, and evaluate its impact on DL models performance, complexity, and trade-offs within the context of cancer classification tasks [6,7].

Furthermore, the study aims to explore the optimization of batch size and image size, which are important parameters in DL model training, to maximize the benefits of stain normalization in less complex models. By finding the optimal combination of these parameters, the study aims to enhance the overall performance of DL models in cancer image classification.

## 2. Materials and Methods

### 2.1. Dataset

This research utilizes two publicly available breast cancer datasets for training the GAN models and evaluating the performance of DL models in multiclass breast cancer classification. The following provides a description of the datasets used:

CAMELYON16 Challenge Dataset was utilized to train the GAN models for stain normalization in two domains: Aperio and Hamamatsu. These two domains represent different imaging scanners commonly used in histopathology. This dataset consists of 400 whole-slide images (WSIs) of sentinel lymph nodes, obtained from two distinct datasets collected at Radboud University Medical Center (Nijmegen, The Netherlands) and the University Medical Center Utrecht (Utrecht, The Netherlands). The training dataset consists of 170 WSIs of lymph nodes, with 100 of them being normal slides and 70 containing metastases [8]. Additionally, there is a second training dataset consisting of 100 WSIs, including 60 normal slides and 40 slides containing metastases. The test dataset consists of 130 WSIs collected from both universities. Figure 1 shows histopathology images from the same stained slide captured using Aperio Scanscope XT scanner and Hamamatsu Nanozoomer 2.0-HT scanner.

ICIAR 2018 Breast Cancer Histology (BACH) Grand Challenge Dataset: This dataset consists of 400 training and 100 test H&E-stained microscopy images with a resolution of 2048 × 1536 pixels. The images were scanned using a Leica DM 2000 LED microscope with a pixel resolution of 0.42 × 0.42 µm. Two expert pathologists labeled the images into four classes. While the labels of training images are available, the labels of test images are withheld [9]. This dataset exhibits significant color variability, making it suitable for color normalization tasks and evaluating the performance of automated cancer diagnostic systems. In this research, the dataset was used for performing multiclass classification of breast histopathology images, specifically classifying them into normal, benign, in situ, and invasive carcinoma classes. Figure 2 shows microscopy images labeled with the predominant cancer type present in each image. The images showcase different cancer types, providing valuable insights into the variations in staining patterns and characteristics within the dataset.

### 2.2. Data Preprocessing

The primary objective of data preprocessing is to convert the original whole-slide images (WSIs) into manageable patch images of sizes 128 × 128, 256 × 256, and 512 × 512, which are suitable for subsequent tasks such as stain normalization and classification. The ICIAR 2018 dataset comprised 2048 × 1536 Tag Image File Format (TIFF) images [10].

For the ICIAR 2018 dataset, the preprocessing phase involved the generation of image patches. This process commenced by applying Otsu thresholding to remove the background from the images, effectively separating the foreground (tissue) from the background and improving the subsequent patch generation process. After the removal of the background, patches were generated at ×40 magnification, resulting in the creation of Portable Network Graphic (PNG) patch images, each with dimensions of 128 × 128, 256 × 256, and 512 × 512 pixels. The generation of multiple patch image sizes aimed to explore the impact of image size variation on subsequent tasks, including stain normalization and classification.

### 2.3. Stain Normalization

Stain normalization in histopathological images aims to standardize the appearance and address color inconsistencies caused by staining protocols, slide preparation techniques, and imaging conditions. This process adjusts the color properties of stained images to achieve uniformity across diverse samples. The evaluation of stain normalization techniques on the performance of DL models for cancer image classification is crucial in enhancing classification performance and developing efficient and accurate systems.

In this study, we employed three specific Generative Adversarial Networks (GANs), namely StainGAN, MultipathGAN, and CycleGAN for the purpose of stain normalization in histopathological images [11,12,13,14]. The utilization of GANs provides several benefits, including the ability to learn intricate mappings between staining protocols, generate realistic normalized images, and enhance the standardization process in cancer image analysis [15,16,17,18,19].

Figure 3 provides a visual representation of the stain normalization results using different GAN models. This visualization offers valuable insights into the impact of these models on enhancing image quality for histopathological analysis. It demonstrates the effectiveness of the GANs in normalizing stained images and improving their visual quality, thereby contributing to more accurate and reliable histopathological analysis.

### 2.4. Generative Adversarial Networks (GANs) Performance Evaluation Metrices

In order to proceed with the performance evaluation of the DL models, we conducted an evaluation of stain normalization processes to select the most suitable stain normalization GAN. This evaluation encompassed assessing the performance of different stain normalization GANs and subsequently evaluating the quality of the generated images. To ensure a comprehensive evaluation, appropriate metrics were employed in this process as follows.

The Structural Similarity Index (SSIM) is a widely used metric for assessing the similarity between two images. It takes into account three components: luminance, contrast, and structure. The SSIM ranges between −1 and 1, where a value of 1 indicates a perfect match [20,21].

The equation for SSIM is as follows:(1)$$SSIM=\frac{\left(2\mu_{x}\mu_{y}+c_{1})(2\sigma_{xy}+c_{2}\right)}{\left(\mu_{x}^{2}+\mu_{y}^{2}+c_{1})(\sigma_{x}^{2}+\sigma_{y}^{2}+c_{2}\right)}$$

In this equation, $\mu_{x}$, $\mu_{y}$, are the mean and $\sigma_{x}$, $\sigma_{y}$ are the standard deviation of the intensity values present in the two images, respectively. $\sigma_{xy}$ is the covariance between the two images’ intensities. The constants $c_{1}$, $c_{2}$ are used for negating the weak denominator effect. 

The Fréchet Inception Distance (FID) is another metric used to evaluate the quality and diversity of generated images. It measures the similarity between the distribution of real images and the distribution of generated images in feature space, as captured by a pre-trained Inception model. A lower FID indicates better image quality and diversity 240 [22]. Mathematically, the Fréchet Distance is used to compute the distance between two “multivariate” normal distribution. For a “univariate” normal distribution, the Fréchet Distance is given as
(2)$$d\left(X,Y\right)={\left(\mu_{X}-\mu_{Y}\right)}^{2}+{\left(\sigma_{X}-\sigma_{Y}\right)}^{2}$$
where *μ* and *σ* are the mean and standard deviation of the normal distributions, *X* and *Y* are two normal distributions.

In the context of GAN evaluation, the FID utilizes feature distances calculated with a pre-trained Inception V3 model. The use of activations from the Inception V3 model to summarize each image gives the FID value.

The Fréchet Inception Distance for “multivariate” normal distribution is given by,
(3)$$FID={\left\|\mu_{X}-\mu_{Y}\right\|}^{2}-Tr\left(\sum X+\sum Y-2\sqrt{\sum X\sum Y}\right)$$
where *X* and *Y* are the real and fake embeddings (activation from the Inception model) assumed to be two multivariate normal distributions. $\mu_{X}$ and $\mu_{Y}$ are the magnitudes of the vector *X* and *Y*. Tr is the trace of the matrix and ∑ *X* and ∑ *Y* are the covariance matrix of the vectors.

The Inception Score (IS) is a metric used to evaluate the quality and diversity of generated images. It measures how well the generated images fool a pre-trained Inception model. A higher IS indicates better image quality and diversity [23].

The equation for IS is as follows:(4)$$IS(G)=exp\left(E_{x\sim pg}D_{KL}\left(p\left.\left(y|X\right)\right\|p\left(y\right)\right)\right)$$
where x~pg indicates that x is an image sampled from pg, $D_{KL}\left(\left.p\right\|q\right)$ is the KL-divergence between the distributions p and q, *p*(*y*|*X*) is the conditional class distribution, and $p\left(y\right)=\int_{X}p(y|X)p_{g}(X)$ is the marginal class distribution.

Additionally, other commonly used metrics for evaluating stain normalization techniques include Mean Squared Error (MSE) and Peak Signal-to-Noise Ratio (PSNR). MSE quantifies the average squared difference between pixel values of the generated and reference images, indicating improved stain normalization with lower MSE values. PSNR measures the ratio between the maximum possible image power and noise power, providing insights into image quality.
(5)$$RMSE=\sqrt{\frac{1}{n_{x}n_{y}}\sum_{i,j}^{n_{x}n_{y}}{\left(\frac{r\left(i,j\right)-t\left(i,j\right)}{r(i,j)}\right)}^{2}}$$

The PSNR computes the peak signal-to-noise ratio, in decibels, between two images. This ratio is used as a quality measurement between the generated and a target image. The higher the PSNR, the better the quality in generated image. The mean square error (MSE) and the peak signal-to-noise ratio (PSNR) are used to compare image quality.
(6)$$PSNR=20{log}_{10}\left(\frac{R^{2}}{{\left(RMSE\right)}^{2}}\right)$$

### 2.5. Image Classification

After generating stain-normalized images using stain normalization GANs, the subsequent step involves performing image classification. In this section, the focus lies in employing deep learning (DL) models to classify stained histopathological cancer images into distinct categories, including benign, in situ, invasive, and normal. The objective is to evaluate the efficacy of the stain normalization techniques in enhancing the classification accuracy of DL models [24,25]. By assessing the performance of DL models on the stain-normalized images, the study aims to determine the impact of stain normalization on the accuracy and reliability of cancer image classification.

The ICIAR2018 Breast Cancer Histology dataset is used for the image classification process. This dataset consists of stained normalized images that have undergone the previously explained stain normalization techniques. The dataset is divided into training, validation, and testing sets to ensure proper evaluation of the models’ performance. It ensures that the same set of images is used for training, validation, and testing, with different DL models [26,27,28]. Six different DL models are employed for image classification, ranging from less complex to high complex DL models including MobileNet, XceptionNet, InceptionV3, ResNet50, VGG16, and VGG19, chosen based on their proven performance in image classification tasks and compatibility with the stained histopathological images [29,30,30,31,32]. Table 1 provides an overview of the DL models used for cancer image classification, including their model size, parameter count, and depth.

In the process of training the DL models, separately stain-unnormalized and stain-normalized image patches were used. Both stain-normalized and stain-unnormalized datasets consisted of image patches of three distinct sizes: 128 × 128, 256 × 256, and 512 × 512. This selection enabled an examination of how different image sizes affected the performance of the DL models when trained on unnormalized datasets.

By incorporating both stain-unnormalized and stain-normalized datasets, the intention was to compare the performance of the DL models on stain-unnormalized images with their performance on images that underwent stain normalization. Through this analysis, the effectiveness of stain normalization in enhancing the models’ classification performance could be assessed. 

Furthermore, the utilization of varying image patch sizes and batch sizes allowed for an evaluation of the impact of image resolution on the performance, efficiency, resource utilization, and trade-offs of the DL models. This facilitated the identification of the optimal image size and batch that yielded the most favorable classification results. By systematically exploring different image resolutions, the study aimed to determine the resolution that strikes the best balance between accuracy and computational efficiency, providing insights into the optimal image size for the classification task.

Figure 4 illustrates the workflow of stain-normalized image classification using a variety of deep learning models. The figure depicts the sequential steps involved in the classification process, highlighting the key stages and interactions between different components.

The evaluation of the image classification results involves analyzing metrics such as accuracy, precision, recall, and F1-score. These metrics provide quantitative measures of the models’ performance in correctly classifying the stained histopathological images into their respective categories. Additionally, the performance of the image classification models is compared with and without the application of stain normalization techniques to assess the impact of the normalization process on classification accuracy.

## 3. Results and Discussion

### 3.1. GAN Models Selection

GANs’ performance evaluation metrics collectively provide quantitative measures to assess the quality, diversity, and visual fidelity of stain-normalized images. By considering both perceptual and statistical aspects, these metrics contribute to a comprehensive assessment of the generated image quality, enabling informed decision-making in stain normalization research. Table 2 presents a comprehensive summary of the evaluation of various stain normalization methods, using a range of quantitative metrics. 

Figure 5 represents the variation of evaluation metrics (SSIM, FID, and IS) with respect to the iteration number for GANs evaluation.

These graphs offer a visual depiction of the dynamic changes in metric values throughout the training process, providing valuable insights into the performance and convergence of the GANs. By observing the trends and fluctuations in the evaluation metrics, researchers can assess the progress and effectiveness of the GAN models and make informed decisions regarding their training and optimization.

The evaluation metrics used in this study provide valuable insights into the performance of each stain normalization method. Through comprehensive analysis, the results clearly indicate that StainGAN exhibits superior performance compared to other stain normalization GANs. The evaluation metrics highlight the effectiveness of StainGAN in achieving accurate and consistent stain normalization, making it a promising choice for enhancing image quality and standardization in histopathological analysis.

### 3.2. Deep Learning Model Performance in Cancer Classification

In the image classification phase, various DL models, including MobileNet, XceptionNet, InceptionV3, ResNet50, VGG16, and VGG19, were employed to classify the stain-normalized histopathological images into different cancer categories. The evaluation of the models’ performance with different image sizes used and involved metrics such as accuracy, precision, recall, and F1-score [8,23,33,34]. 

Table 3 summarizes the classification performance of different DL models on the dataset with varying image sizes (128 × 128, 256 × 256, and 512 × 512) without stain normalization. The primary objective of this table is to demonstrate the models’ effectiveness in cancer classification when stain normalization is not employed. 

When analyzing the performance metrices, it is evident that VGG16 consistently achieved the highest accuracy scores across all image sizes. Specifically, it attained an accuracy of 68.22% on the 128 × 128 image size, 71.59% on the 256 × 256 image size, and 76.80% on the 512 × 512 image size. Similarly, among the models examined, Xception demonstrated strong performance, particularly excelling on the 256 × 256 and 512 × 512 image sizes with an accuracy of 68.92% and 75.25%, respectively. 

In contrast, ResNet50 exhibited comparatively lower accuracy scores across all image sizes, suggesting a relatively weaker performance on this unstain-normalized dataset. These findings emphasize the influence of image size on the models’ classification performance. Notably, larger image resolutions, such as the 512 × 512 size, tend to yield higher accuracy scores, potentially augmenting the models’ ability to discern subtle patterns and features within the histopathological images.

Table 4 summarizes the performance of different DL models on the dataset with varying image sizes (128 × 128, 256 × 256, and 512 × 512) with stain normalization. The primary objective of this table is to demonstrate the models’ effectiveness in cancer classification when stain normalization is employed.

The analysis of the results from the provided Table 4 reveals significant insights into the performance of DL models trained on stain-normalized datasets at different image sizes. Across all image sizes, VGG16 consistently achieved the highest accuracy scores on the stain-normalized dataset. It obtained accuracy values of 74.24% for the 128 × 128 image size, 85.29% for the 256 × 256 image size, and 88.64% for the 512 × 512 image size. These results demonstrate the robustness of VGG16 in accurately classifying cancer samples when stain normalization is applied. The XceptionNet exhibited strong performance on the stain-normalized dataset, particularly excelling on the 256 × 256 and 512 × 512 image sizes with accuracies of 82.92% and 85.13%, respectively. This suggests that XceptionNet is effective in capturing relevant features and patterns in histopathological images, even after stain normalization.

In contrast, ResNet50 showed relatively lower accuracy scores across all image sizes on the stain-normalized dataset, indicating its comparatively weaker performance in this context. This suggests that ResNet50 might struggle to fully exploit the benefits of stain normalization for improving classification accuracy. Examining precision, recall, and F1-score, VGG16 consistently achieved high scores across all image sizes on the stain-normalized dataset. This demonstrates that VGG16 not only achieved high accuracy but also exhibited a good balance between true positives and false positives, resulting in high precision and recall values.

The results emphasize the influence of image size on the models’ performance, when stain normalization is applied. Larger image resolutions, such as the 512 × 512 size, tend to yield higher accuracy scores, indicating the potential enhancement in capturing subtle patterns and features within stain-normalized histopathological images.

The analysis of the results presented in Table 3 and Table 4 unveiled a notable enhancement in the classification accuracy of the DL models upon the application of stain normalization techniques. The incorporation of stain-normalized images, which were generated using stain normalization GANs, ensured a consistent visual depiction of tissue structures across diverse samples. This consistency, in terms of color and intensity, played a crucial role in enabling the DL models to extract more meaningful features. Consequently, the models exhibited improved accuracy in cancer classification tasks. These findings emphasize the effectiveness of stain normalization in standardizing the image data and enhancing the models’ capacity to discern relevant patterns and structures, thus leading to more accurate classification outcomes.

### 3.3. Computational Complexity Analysis

The computational complexity analysis aimed to investigate the impact of input image sizes and batch sizes on the resource utilization of DL models used in cancer image classification. The analysis involved a comprehensive comparison of various performance metrics, including the number of parameters and image size in the DL models, processing speed in relation to both image size and batch size, FLOPs (floating-point operations per second) relative to image size, and the correlation between image size and batch size with GPU usage. This evaluation encompassed diverse DL models, each employing different input image sizes and batch sizes.

Table 5 provides information on different models, including their respective image sizes, number of parameters, and FLOPs (floating-point operations) measured in millions. The FLOPs served as a measure of computational complexity, providing insights into the computational demands of the models. 

Through meticulous analysis of these performance metrics, this investigation yielded invaluable insights into the trade-offs, complexities, and resource requirements associated with DL models deployed in breast cancer image classification tasks that encompass diverse input image sizes and batch sizes. 

The experiments were conducted using a computer system with specific specifications to ensure efficient execution of the DL experiments. The computer system used for these experiments was equipped with an Intel Core i5 processor running at 3.5 GHz, 64 GB of RAM, and a high-performance NVIDIA GeForce RTX 4090 graphics card.

The choice of this computer system was driven by the need for substantial computational power to handle the large-scale DL tasks involved in training and evaluating the models. The inclusion of the NVIDIA GeForce RTX 4090 graphics card ensured accelerated training and inference processes, leveraging the card’s parallel computing capabilities.

Figure 6 illustrates the relationship between the number of FLOPs and the number of parameters in different DL models. The FLOPs metric provides insights into the computational complexity of the models, reflecting the number of arithmetic operations required for processing the input data. 

In Figure 6, the size of the plot demonstrates size of input image and how the number of FLOPs changes as the number of parameters varies across different models. Each point on the graph represents a specific model configuration, with the x-axis denoting the number of parameters and the y-axis representing the corresponding number of FLOPs.

In the case of the models MobileNet, Xception, InceptionV3, and ResNet50, the number of training parameters remained constant regardless of the input image size. However, the number of floating-point operations (FLOPs) performed during model inference varied based on the input image size. This means that as the size of the input image increased, the computational workload in terms of FLOPs also increased. This insight is valuable for optimizing computational efficiency and resource allocation when utilizing these models, as it allows for better understanding of the computational requirements associated with different input sizes.

To evaluate the effect of increased image sizes on classification performance, the accuracy of each model was measured using the test dataset. Additionally, the processing speed, quantified as the number of images processed per second (IPS), was examined to identify disparities in computational efficiency. The findings of this investigation are summarized in Table 6, offering a comprehensive overview of diverse DL models. The table presents relevant information such as image sizes, the number of images processed per second, and corresponding batch and images sizes. By conducting a comparative assessment of processing speeds across different image sizes and batch sizes, valuable insights were obtained concerning the computational efficiency and capability of each model to handle varying workloads.

The results presented in Table 3 and Table 4 provide evidence that increasing the image size has a positive impact on cancer classification performance of the DL models. The larger image sizes allow for capturing more detailed information, leading to improved accuracy in classification. However, it is important to consider the potential challenges associated with increasing image size, such as computational complexity and increased training and inference time.

To further investigate the impact on processing speed, we examined the relationship between batch size and image size on DL model performance. We explored how varying these parameters influenced the computational demands of the models during training and inference. By analyzing the processing speed, we aimed to identify an optimal balance between image size and batch size that maximizes both classification accuracy and computational efficiency.

Table 6 provides information about the processing speed of different DL models measured in IPS for various image sizes (128 × 128, 256 × 256, 512 × 512) and batch sizes (1, 2, 4, 8, 16, 32, 64, 128, 256). 

Figure 7 illustrates the relationship between the number of IPS and the batch size in various DL models when image size is changed. The IPS metric serves as a valuable indicator of the computational efficiency of the models, quantifying the number of images processed within a second across different image sizes and batch sizes.

Analyzing the results, it is evident that increasing the image size generally leads to a decrease in processing speed across all models. This is expected since larger images contain more pixels and require more computational resources, resulting in a reduced number of images processed per second.

Furthermore, Table 6 shows that the batch size also influences the processing speed. Generally, as the batch size increases, the processing speed improves, indicating better utilization of parallel processing capabilities. However, there is a diminishing return in speed improvements beyond a certain batch size, as the models may experience limitations in memory or computational capacity. 

Considering specific models, it can be observed that MobileNet generally achieves the highest processing speeds compared to other models, particularly at larger image sizes and batch size. On the other hand, Inception3 also consistently demonstrates faster processing speeds across different image sizes and batch sizes.

This relationship provides meaningful insights into the models’ ability to handle larger workloads and deliver faster processing speeds, which are crucial considerations for optimizing DL model performance in real-world applications.

Furthermore, our analysis included an examination of memory utilization to investigate the impact of larger input image sizes and varying batch sizes on the utilization of Graphics Processing Unit (GPU) memory. Table 7 provides insights into GPU memory utilization in DL models, highlighting the impact of image and batch sizes on resource demands.

Among the DL models analyzed, MobileNet consistently demonstrates lower GPU memory requirements compared to others, irrespective of image size or batch size. Xception and InceptionV3 also exhibit relatively low GPU memory usage, with InceptionV3 occasionally showing slightly higher requirements than Xception. ResNet50 generally demands more GPU memory than MobileNet, Xception, and InceptionV3. Notably, VGG16 and VGG19 exhibit the highest GPU memory usage across the analyzed models, even with smaller image sizes and batch sizes. These findings emphasize the importance of considering GPU memory limitations when selecting a DL model, as higher memory requirements may restrict the feasible batch size or image size. Employing optimization techniques such as model pruning can be beneficial in reducing the GPU memory footprint. 

A subset of data corresponding to specific batch sizes is absent in the 256 × 256 and 512 × 512 image resolutions for certain DL models. This discrepancy arises from the inherent limitations of computer GPU memory. The constrained memory capacity of the GPU hindered the feasibility of processing and storing the entire dataset for these batch sizes at the aforementioned higher image resolutions. Consequently, the absence of data points in the experimental results directly stems from these hardware limitations.

Figure 8 illustrates the relationship between the number of GPU memory utilization and the batch size in different DL models when image size is changed. The GPU memory utilization metric serves as a valuable indicator of the computational resource demand of the models, quantifying the amount of GPU memory used during the training of DL models for different image sizes and batch sizes. 

The absence of data for certain batch sizes of DL models in the 256 × 256 and 512 × 512 image sizes is attributed to the inability to complete the processing task due to insufficient GPU memory. These larger image sizes necessitate a substantial amount of memory for processing, and when the allocated GPU memory falls short, the task cannot be executed successfully. This limitation arises from the inherent physical limitations of the GPU memory capacity, which imposes restrictions on the sizes of images that can be processed. As a result, data collection and analysis for batch sizes in these image sizes were not possible due to the impracticality of storing and manipulating the necessary data within the available memory resources. This underscores the critical importance of effective memory resource management and considering the hardware limitations when working with DL models that involve large image sizes and batch sizes. Employing memory optimization techniques and adopting memory-efficient architectures can help alleviate these constraints and enable the processing of larger image sizes within the constraints of the available GPU memory.

These insights played a crucial role in identifying potential challenges and limitations related to GPU memory capacity, providing invaluable guidance for optimizing the selection of image sizes and batch sizes. The ultimate goal of these optimization efforts was to maximize the effective utilization of available computational resources and ensure the smooth execution of the DL models. This approach aimed to strike a balance between resource efficiency and computational performance, allowing for efficient utilization of the GPU and facilitating optimal model performance.

## 4. Discussion

The findings of our study highlight the effectiveness of DL models for cancer classification and the importance of stain normalization techniques in improving their performance. Our evaluation revealed that VGG16 emerged as the best model for cancer classification, thanks to its deep architecture and large number of parameters. However, it is worth noting that other models such as MobileNet and Xception can also deliver competitive performance when stain normalization techniques are properly applied.

Stain normalization plays a critical role in addressing the challenge of staining variations. These stain variations can introduce image appearance discrepancies and hinder accurate classification. By applying H&E stain normalization on histopathology images, these variations can be mitigated, allowing the models to focus on relevant features and patterns. The significance of stain normalization is particularly pronounced for less complex models such as MobileNet, Xception, and Inception, as their fewer parameters may limit their ability to capture subtle differences caused by staining variations.

Our results indicate that stain normalization greatly impacts the performance of these less complex models, enabling them to achieve improved accuracy and efficiency in stain normalized cancer image classification tasks. This finding emphasizes the need to consider both the model architecture and the implementation of stain normalization techniques when developing cancer classification models. By incorporating stain normalization into the workflow, researchers and developers can enhance the performance of less complex models and achieve results comparable to more complex models such as VGG16.

Furthermore, the efficiency and computational complexity of MobileNet and Xception make them attractive candidates for practical applications. These models offer a good balance between performance and resource requirements, making them suitable for deployment in real-world scenarios where computational resources may be limited.

Although our evaluation showcased the impressive performance of various DL models for cancer classification, it is important to acknowledge the limitations and failure of certain models. In our study, the InceptionV3, ResNet50, and VGG19 models exhibited subpar performance compared to other models, highlighting its failure in accurately classifying cancer images.

VGG19, an extension of the VGG16 model, features a deeper architecture with 19 layers. While the increased depth allows for more complex representations to be learned, it also introduces challenges such as vanishing gradients and increased computational requirements. In our study, we found that despite its increased depth, VGG19 did not yield superior performance compared to VGG16. This suggests that the additional layers may not have provided substantial benefits for cancer classification, potentially due to diminishing returns or the risk of overfitting the data.

ResNet50, on the other hand, is a popular residual network architecture that addresses the vanishing gradient problem by introducing residual connections. These connections enable the gradient to flow directly from earlier layers to later layers, facilitating the training of deeper networks. However, in our experiments, ResNet50 did not perform as well as some of the other models, including VGG16. This could be attributed to several factors, such as the complexity of cancer images and the specific dataset used in our study. It is possible that ResNet50’s architecture may not have been optimized for capturing the relevant features and patterns in stain-normalized cancer images.

The low performance of complex DL models such as InceptionV3 and ResNet50 in our study highlight the importance of model selection and the need for careful consideration when choosing an appropriate architecture for stain-normalized cancer image classification. It is crucial to assess the suitability of a model’s architecture, complexity, efficiency, and ability to capture relevant features within the context of the specific dataset and task at hand. While InceptionV3 and ResNet50 may have demonstrated strong performance in other domains or with different datasets, their performance limitations in our study underscore the need for thorough evaluation and model selection in the context of stain-normalized cancer classification.

## 5. Conclusions

In conclusion, our study highlights the significant impact of H&E stain normalization on the selection of DL models for cancer image classification. While VGG16 exhibited strong performance, InceptionV3 and ResNet50 faced limitations in this context. Notably, stain normalization techniques greatly enhanced the performance of less complex models such as MobileNet, Xception, and Inception. These models emerged as competitive alternatives with lower computational complexity, improved computational efficiency, and reduced resource requirements. The findings underscore the importance of optimizing less complex models through stain normalization, achieving accurate and reliable cancer image classification while striking a balance between performance, complexity, efficiency, and trade-offs. This research holds tremendous potential for advancing the development of computationally efficient cancer classification systems.

## Figures and Tables

**Figure 1 cancers-15-04144-f001:**
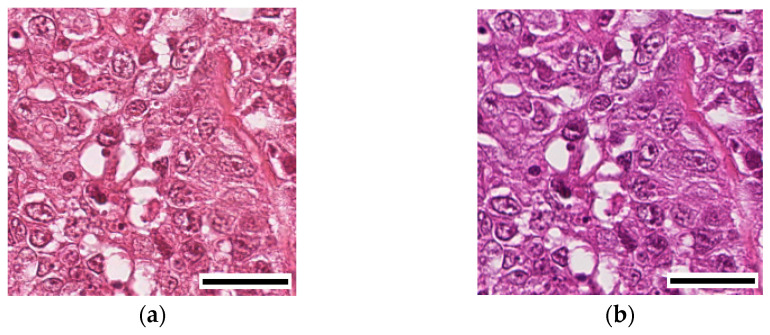
(**a**) Image acquired using Aperio Scanscope XT scanner. (**b**) Image acquired using Hamamatsu Nanozoomer 2.0-HT scanner. Scale bar, 16.95 µm.

**Figure 2 cancers-15-04144-f002:**

Microscopy images labeled with the predominant cancer type in each image: (**a**) Normal, (**b**) benign, (**c**) in situ carcinoma, and (**d**) invasive carcinoma. Scale bar, 164 µm.

**Figure 3 cancers-15-04144-f003:**
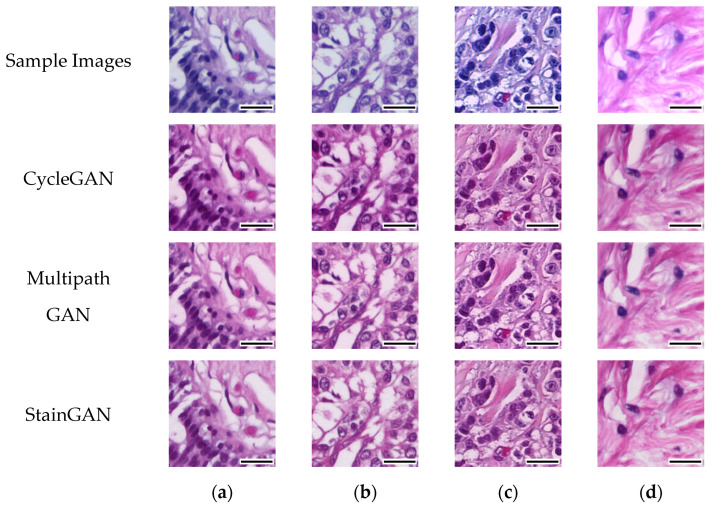
CycleGAN, MultipathGAN, and StainGAN normalized sample images from the dataset, labeled with the predominant cancer type in each image. (**a**) Benign, (**b**) in situ carcinoma, (**c**) invasive carcinoma, and (**d**) normal. Scale bar, 31.5 µm.

**Figure 4 cancers-15-04144-f004:**
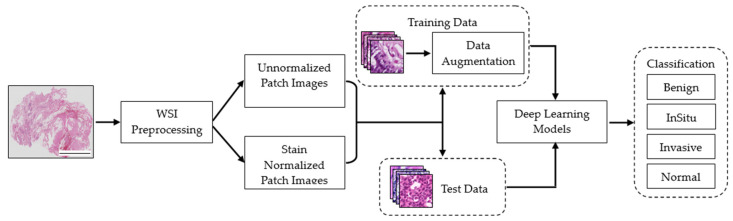
Workflow of stain normalized images classification using diverse deep learning models. Scale bar, 11,698.35 µm.

**Figure 5 cancers-15-04144-f005:**
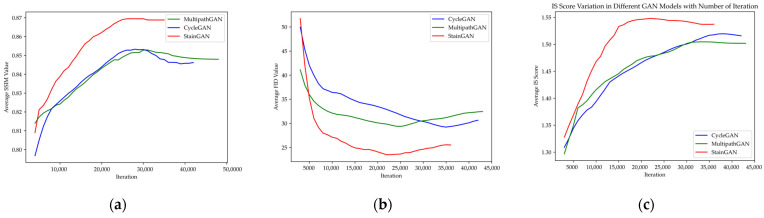
Variation of evaluation metrics (SSIM, FID, and IS) with respect to the iteration number for GANs evaluation. (**a**) Variation of SSIM values; (**b**) variation of FID values; (**c**) variation of IS score.

**Figure 6 cancers-15-04144-f006:**
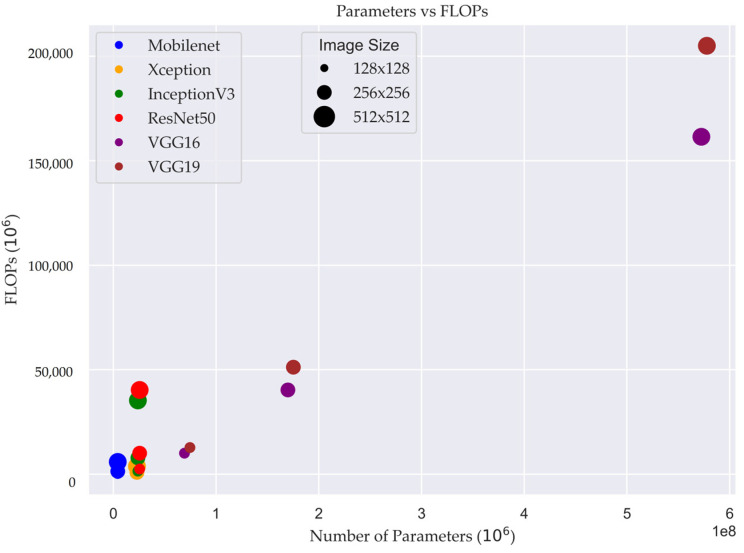
Relationship between model complexity and computational efficiency in deep learning: FLOPs versus parameters.

**Figure 7 cancers-15-04144-f007:**
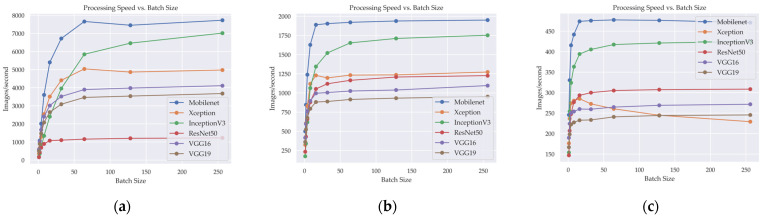
Relationship between model computational efficiency and batch size in different input image sizes in deep learning models: (**a**) Image size 128 × 128, (**b**) image size 256 × 256, and (**c**) image size 512 × 512.

**Figure 8 cancers-15-04144-f008:**
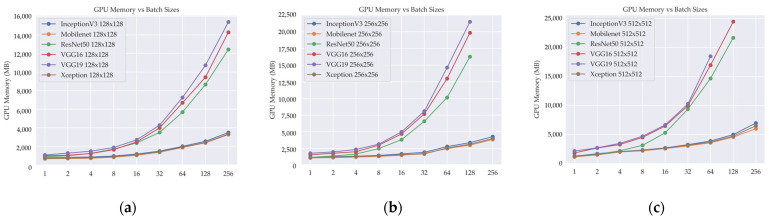
Relationship between GPU memory utilization and batch size in different input image sizes in deep learning models: (**a**) image size 128 × 128, (**b**) image size 256 × 256, and (**c**) image size 512 × 512.

**Table 1 cancers-15-04144-t001:** Deep learning models used for cancer image classification.

Model	Size (MB)	Number of Parameters	Model Depth
MobileNet	16	4.3 M	55
MobileNetV2	14	3.5 M	105
XceptionNet	88	22.9 M	81
InceptionV3	92	23.9 M	189
ResNet50	98	25.6 M	107
VGG16	528	138.4 M	16
VGG19	549	143.7 M	19

**Table 2 cancers-15-04144-t002:** Comparative evaluation of stain normalization methods based on image quality metrics.

Stain Normalization GANs	SSIM	FID	IS	PSNR	RMSE
StainGAN	0.868 ± 0.024	23.417 ± 3.135	1.549 ± 0.064	15.805 ± 0.862	42.604 ± 4.671
MultipathGAN	0.854 ± 0.034	29.135 ± 5.434	1.505 ± 0.102	15.548 ± 2.031	45.186 ± 9.409
CycleGAN	0.844 ± 0.036	29.291 ± 5.156	1.519 ± 0.114	15.357 ± 1.883	44.426 ± 8.519

**Table 3 cancers-15-04144-t003:** Classification performance of deep learning models on dataset without stain normalization and with different image sizes.

Image Size	Deep Learning Model	Accuracy	Precision	Recall	F1-Score
128 × 128	MobileNet	0.6269	0.6487	0.6269	0.6220
	Xception	0.6267	0.6287	0.6248	0.6254
	InceptionV3	0.6121	0.6280	0.6122	0.6112
	ResNet50	0.5657	0.5725	0.5658	0.5664
	VGG16	0.6822	0.7003	0.6822	0.6838
	VGG19	0.5784	0.5832	0.5784	0.5787
256 × 256	MobileNet	0.6527	0.7113	0.6560	0.6429
	Xception	0.6892	0.6889	0.6892	0.6881
	InceptionV3	0.6268	0.6486	0.6269	0.6220
	ResNet50	0.5983	0.6203	0.5984	0.5954
	VGG16	0.7159	0.7318	0.7159	0.7168
	VGG19	0.6607	0.7317	0.6664	0.6515
512 × 512	MobileNet	0.7131	0.7414	0.7143	0.7135
	Xception	0.7525	0.7571	0.7525	0.7535
	InceptionV3	0.6877	0.6894	0.6861	0.6862
	ResNet50	0.6156	0.6162	0.6150	0.6155
	VGG16	0.7680	0.7867	0.7680	0.7681
	VGG19	0.7584	0.7666	0.7584	0.7599

**Table 4 cancers-15-04144-t004:** Classification performance of deep learning models on dataset with stain normalization and with different image sizes.

Image Size	Deep Learning Model	Accuracy	Precision	Recall	F1-Score
128 × 128	MobileNet	0.6741	0.6767	0.6741	0.6733
	Xception	0.6805	0.7756	0.6847	0.6829
	InceptionV3	0.6643	0.6688	0.6635	0.6652
	ResNet50	0.6146	0.6158	0.6146	0.6149
	VGG16	0.7424	0.7699	0.7424	0.7382
	VGG19	0.6587	0.6676	0.6587	0.6594
256 × 256	MobileNet	0.8277	0.8357	0.8277	0.8267
	Xception	0.8292	0.8411	0.8292	0.8310
	InceptionV3	0.7669	0.7726	0.7669	0.7678
	ResNet50	0.7199	0.7311	0.7199	0.7206
	VGG16	0.8529	0.8618	0.8529	0.8536
	VGG19	0.8072	0.8232	0.8072	0.8083
512 × 512	MobileNet	0.8672	0.8722	0.8672	0.8652
	Xception	0.8513	0.8632	0.8512	0.8531
	InceptionV3	0.7637	0.7846	0.7638	0.7632
	ResNet50	0.7339	0.7466	0.7339	0.7345
	VGG16	0.8864	0.8869	0.8864	0.8862
	VGG19	0.8130	0.8138	0.8130	0.8121

**Table 5 cancers-15-04144-t005:** Comparison of deep learning models: image size, number of parameters, and FLOPs.

Model Name	Image Size	Number of Parameters (×10^6^)	FLOPs (×10^6^)
MobileNet	128 × 128	4.23	372.97
	256 × 256	4.23	1485.43
	512 × 512	4.23	5934.96
Xception	128 × 128	22.86	239.03
	256 × 256	22.86	958.55
	512 × 512	22.86	3851.63
InceptionV3	128 × 128	23.82	1528.10
	256 × 256	23.82	7848.18
	512 × 512	23.82	35,399.45
ResNet50	128 × 128	25.58	2532.11
	256 × 256	25.58	10,101.11
	512 × 512	25.58	40,362.60
VGG16	128 × 128	69.15	10,133.14
	256 × 256	169.81	40,407.33
	512 × 512	572.47	161,504.10
VGG19	128 × 128	74.46	12,851.05
	256 × 256	175.12	51,278.97
	512 × 512	577.78	204,990.64

**Table 6 cancers-15-04144-t006:** Processing speed (images per second) of different image and batch sizes in deep learning models.

Model Name	Image Size	Batch Size								
		1	2	4	8	16	32	64	128	256
MobileNet	128 × 128	540.15	1081.93	2015.89	3603.89	5401.05	6719.05	7660.02	7451.87	7728.02
	256 × 256	499.39	845.63	1239.18	1627.30	1889.37	1904.56	1921.14	1938.62	1950.49
	512 × 512	245.67	330.38	415.58	441.72	473.93	475.73	477.47	476.42	470.84
Xception	128 × 128	350.60	677.29	1283.17	2542.02	3508.50	4409.72	5038.36	4865.04	4978.35
	256 × 256	323.11	606.33	830.59	1119.34	1228.95	1196.50	1232.76	1235.98	1272.97
	512 × 512	176.37	236.75	273.99	280.28	285.12	272.57	260.54	244.76	229.11
InceptionV3	128 × 128	181.49	360.02	699.41	1340.37	2401.51	3956.92	5846.22	6462.29	7017.33
	256 × 256	174.06	340.00	618.27	1063.91	1344.54	1521.71	1653.22	1710.76	1752.93
	512 × 512	153.89	254.46	324.17	363.19	394.28	405.36	417.29	421.13	424.68
ResNet50	128 × 128	158.43	430.28	686.12	897.26	1076.86	1101.83	1156.99	1202.26	1225.30
	256 × 256	233.74	429.29	657.80	880.03	1052.60	1119.81	1164.72	1207.70	-
	512 × 512	146.97	207.08	247.62	276.85	293.37	299.92	305.13	307.33	-
VGG16	128 × 128	588.77	1015.01	1674.91	2399.89	3011.98	3516.27	3896.29	3979.16	4111.39
	256 × 256	416.60	595.79	761.39	895.84	995.87	1004.95	1025.18	1038.52	-
	512 × 512	189.76	224.26	250.75	254.00	259.98	259.52	264.67	269.04	-
VGG19	128 × 128	497.74	894.28	1472.74	2077.66	2649.24	3087.24	3462.04	3537.24	3676.34
	256 × 256	361.73	524.49	676.96	794.41	881.99	888.68	915.30	932.38	-
	512 × 512	166.89	198.35	222.40	227.34	232.69	233.35	240.80	-	-

**Table 7 cancers-15-04144-t007:** GPU memory utilization in different deep learning models with image and batch sizes.

Model Name	Image Size	Batch Size								
		1	2	4	8	16	32	64	128	256
MobileNet	128 × 128	654.72	686.84	714.36	824.92	1032.46	1362.24	1874.98	2368.62	3264.04
	256 × 256	1096.16	1136.68	1208.65	1312.42	1492.93	1772.06	2483.44	2994.19	3846.82
	512 × 512	1134.66	1486.16	1992.82	2192.60	2572.84	3024.18	3582.36	4548.02	5994.56
Xception	128 × 128	698.00	718.26	764.82	866.90	1098.16	1408.38	1896.30	2414.00	3316.94
	256 × 256	1121.04	1208.32	1297.45	1386.82	1568.10	1689.06	2578.16	3145.26	3994.14
	512 × 512	1206.81	1538.72	2008.46	2264.82	2612.00	3165.78	3685.70	4726.12	6492.24
InceptionV3	128 × 128	758.14	796.52	846.28	948.92	1164.18	1476.82	1992.46	2536.28	3476.21
	256 × 256	1184.34	1268.68	1352.25	1462.00	1685.10	1926.56	2786.41	3369.72	4279.15
	512 × 512	1276.46	1684.82	2048.60	2314.00	2688.84	3246.21	3872.65	4981.09	6974.26
ResNet50	128 × 128	862.54	986.14	1236.82	1692.74	2362.68	3494.82	5676.46	8648.94	12,426.17
	256 × 256	1162.88	1368.62	1668.48	2486.84	3822.62	6574.48	10136.83	16264.43	-
	512 × 512	1326.76	1578.56	2186.26	3124.33	5294.14	9364.69	14,664.14	21,672.97	-
VGG16	128 × 128	982.46	1032.42	1216.94	1635.54	2468.26	3981.46	6686.72	9436.84	14,281.27
	256 × 256	1562.25	1784.19	2014.73	2986.29	4686.24	7682.42	12984.61	19842.47	-
	512 × 512	1824.64	2662.73	3264.48	4496.12	6462.62	9962.63	16,962.78	24,462.92	-
VGG19	128 × 128	1062.66	1263.78	1469.93	1865.03	2694.04	4268.82	7252.92	10,721.82	15,386.22
	256 × 256	1782.42	1984.82	2348.56	3146.94	4984.56	8086.48	14,628.94	21,485.39	-
	512 × 512	2142.36	2696.42	3468.76	4686.62	6662.64	10,264.53	18,492.62	-	-

## Data Availability

The publicly shared datasets are available at https://camelyon17.grand-challenge.org/ (accessed on 22 February 2023), https://iciar2018-challenge.grand-challenge.org/Dataset/ (accessed on 16 March 2023). The codes are available from the author (N.M.; nuwanmadusanka@hotmail.com) upon reasonable request.

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
