# Peer review of "Impact of H&E Stain Normalization on Deep Learning Models in Cancer Image Classification: Performance, Complexity, and Trade-Offs"

_cancers, 2023, doi:10.3390/cancers15164144_

Round 1

Reviewer 1 Report

The article seems interesting and the results carried out we're good enough to support the title of the article, but even the. The article needs to propose somethings with atleast changes in the layers or activation functions of the transfer learning methods no innovativeness is found in the article . 

Author Response

We sincerely appreciate your thoughtful evaluation of our manuscript titled "Impact of H&E Stain Normalization on Deep Learning Models in Cancer Image Classification: Performance, Complexity, and Trade-Offs." Your positive comments regarding the article's interest and the adequacy of the results to support the title are truly encouraging to us.

We understand your suggestion regarding proposing changes to the layers or activation functions of the transfer learning methods to introduce innovativeness into the article. However, we would like to emphasize that the primary aim of our research was to evaluate the specific impacts of H&E Stain Normalization on Deep Learning Models in Cancer image classification. To ensure a clear understanding of these impacts, we deliberately avoided introducing architectural changes to well-known models.

The rationale behind this decision lies in the potential confounding effect that architectural changes could introduce. By focusing solely on the effect of stain normalization, we aimed to provide a comprehensive and unambiguous assessment of its impact on deep learning models' performance, complexity, and trade-offs.

Introducing alterations to widely used architectures could lead to performance improvements, but it would be challenging to attribute those enhancements specifically to stain normalization or architectural changes. Our intention was to isolate the impact of H&E Stain Normalization, thus providing a clearer picture of its significance in cancer image classification.

In light of your feedback, we will consider discussing potential avenues for future research, where architectural modifications and other innovative approaches could be explored in conjunction with stain normalization to further advance the field of cancer image classification.

Once again, we extend our gratitude for your valuable feedback, and we hope that our clarification on the methodology and scope of our study addresses your concerns. We are committed to improving the quality and significance of our research, and your input plays a crucial role in achieving that goal.

Thank you for your time and consideration.

Reviewer 2 Report

Generally speaking, this research addresses an interesting and practical issue that would likely be of interest to the community. The study investigates the impact of H&E stain normalization on the performance of deep learning models in cancer image classification. The authors also conducted a thorough evaluation of the performance of well-known networks such as VGG19, VGG16, ResNet50, MobileNet, Xception, and InceptionV3 on well-known datasets of H&E stained cancer images. The incorporation of GANs in the analysis is a noteworthy aspect of the research.

However, I have concerns about the significance of considering batch size and image size in this particular context. I believe the authors should provide further clarification to justify the generalization of their findings. Additionally, I recommend that the authors address some typos and grammar issues present throughout the paper. Despite these concerns, the manuscript is clear and readable. I will now elaborate on my concerns in more detail.

-         Let's begin with batch size optimization. While setting a proper batch size could be of importance in deep learning-based image classification, I believe it may not be that crucial for optimizing the model's performance to be discussed in the paper besides the other more important factor that is the impact of stain normalization. I mean, while batch size affects the training dynamics and memory requirements, it is much less important than many other parameters and factors such as the network architecture itself as well as learning rate, regularization techniques, and data augmentation, that play more crucial roles. Although the authors attempted to quantify the significance of this parameter, it remains uncertain whether the same or a similar effect would apply to other datasets beyond those used in this research paper. Therefore, a clear justification is necessary in order to establish the generalizability of these findings.

-         Following my previous comment, I am not still against considering the effect of batch size on the classification performance. However, I really doubt if the effect of image size (or let’s say patch size) is actually worth being considered as one of the important parameters to be optimized for a proper classification. Indeed, most of the DL networks allow the user to use patching while training. Generally speaking, the choice of patch size in deep learning-based image classification should consider the balance between capturing relevant spatial information, object size and scale, computational resources, available training data, generalization, and the input resolution in terms of pixel size. Then let’s say all these factors rather than the image size. Therefore, again, I am not convinced that considering these two factors (i.e., batch size and image size) would be of significant importance in this context.

-         I suggest conducting a thorough proofreading of the manuscript to address various issues present throughout.

Minor remarks:

-         In Figure five, only one of the sub-figures (the one on the right) has a caption above it. I recommend either adding captions to the other two sub-figures or removing the caption for all sub-figures to ensure consistency.

-         In Table 5, the authors provided a useful comparison of deep learning models in terms of the number of parameters and FLOPs. Specifically, the FLOPs metric provides insights into the computational complexity of the models. However, it would be even more beneficial if they could also report the computational time. Including the computational time would make the comparison more meaningful for potential users of these models in the same or similar applications.

-         I would suggest adding references next to each model name in Table 5. This would allow readers to easily locate relevant papers associated with each model.

I recommend that the authors address some typos and grammar issues present throughout the paper. Despite these concerns, the manuscript is clear and readable. 

Author Response

Reply Reviewer’s Comments 1:
Thank you for your valuable feedback on batch size optimization. We acknowledge the significance of factors such as stain normalization, network architecture, learning rate, regularization techniques, and data augmentation in optimizing model performance. Our research focuses on evaluating the Impact of H&E Stain Normalization on Deep Learning Models in Cancer Image Classification: Performance, Complexity, and Trade-Offs. We deliberately avoided introducing architectural changes to ensure a clear assessment, as these changes could confound the effects, making it challenging to attribute improvements solely to stain normalization. Additionally, our research provides a better understanding of the impact of image size and batch size on processing speed (Table 6), as well as the resource requirements and limitations of DL models under available computational resources.

While batch size may not directly impact model accuracy, it is crucial for the following reasons:

1. Training Dynamics: Batch size directly influences training dynamics, affecting parameter updates, convergence, and generalization.
2. Memory Requirements: Batch size affects memory usage, especially on resource-constrained hardware like GPUs.
3. Generalization: Proper batch size selection can improve model generalization, especially with smaller batch sizes introducing more stochasticity.
4. Training Time: Larger batch sizes expedite training due to increased parallelism.

Regarding generalizability, we agree it's essential, but certain aspects like Training Dynamics, Memory Requirements, and Training Time will remain consistent across datasets if using the same batch size and image size.

We maintain that batch size optimization is crucial due to its impact on training dynamics, memory usage, generalization, and training time. Our aim is to provide a comprehensive understanding of the impact of stain normalization on cancer image classification without confounding variables. We appreciate your feedback and will ensure the paper conveys these points clearly.

Reply Reviewer’s Comments 2:
Thank you for sharing your insights on batch size and image size in our classification performance. Table 3 and Table 4 in our research clearly demonstrate how classification model accuracy improved with image size in both stain-normalized and non-stain normalized data for the same dataset. These results underscore the importance of image size in achieving better classification performance and its potential impact on model accuracy. However, we observed a trade-off in computational resource requirements when trying to improve image classification accuracy (refer to Table 8 and Figure 7) via increasing image size. MobileNet, Xception, and InceptionV3 models exhibit low computational resource requirements and high processing speeds.

Optimizing image size allows us to capture critical features and spatial information necessary for accurate classification. Larger image sizes preserve fine-grained details and contextual information, enhancing the model's ability to discriminate between different classes. However, it comes with the downside of increased computational resources, including memory and processing power, as well as processing time.

While we acknowledge your thoughts, we firmly believe that considering both batch size and image size is essential for a comprehensive approach to deep learning-based image classification. These factors collectively contribute to the overall performance and generalization capabilities of the model.

Our research aims to analyze various parameters related to the dataset (not with the DL models) impacting classification performance, including batch size, image size, and stain normalization. By addressing these aspects, we provide a more comprehensive understanding of the factors influencing model accuracy and make informed decisions when deploying deep learning models for cancer image classification.

We appreciate your feedback, and we hope that Table 3 and Table 4 clarify the significance of image size optimization in our study. Your valuable input has been instrumental in improving the quality of our work.

Reply Reviewer’s Comments 3:
Thank you for your feedback and suggestion. We appreciate your attention to detail, and we agree that thorough proofreading of the manuscript is essential to ensure the overall quality and clarity of the content.

We will carefully review the manuscript and address all the issues you have pointed out. Our goal is to enhance the readability and coherence of the paper, ensuring that it effectively communicates our research findings and insights to the readers.

Reply Reviewer’s Comments 4:
We will take immediate action to address this inconsistency and ensure that all sub-figures within Figure five have appropriate captions. Alternatively, if removing captions from all sub-figures results in a more consistent presentation, we will make that adjustment accordingly.

Reply Reviewer’s Comments 5:
You are absolutely right, and we understand the importance of providing insights into the computational time of these models. In our research, we have addressed this aspect in Table 6, which provides similar information but in terms of computational speed. Table 6 shows the number of images processed per second in different DL models with varying image sizes and batch sizes.

By including the processing speed information, we aim to offer a comprehensive understanding of computational efficiency. This information is essential for potential users who need to consider both model performance and computational resources when choosing the most suitable deep-learning model for their applications.

Reply Reviewer’s Comments 6:
We appreciate your attention to detail and believe that this enhancement will further improve the clarity and usability of our manuscript. We will promptly update Table 5 to include the appropriate references for each deep learning model mentioned.

Thank you once again for your valuable feedback and suggestions. Your input has been instrumental in refining our research, and we are committed to incorporating these improvements into the revised manuscript. If you have any other suggestions or areas of interest, please feel free to share, as we highly value your feedback in further enhancing the quality and impact of our work.

Reviewer 3 Report

Overall, this research will be interesting for the field of cancer diagnosis with HE images, but there are still some issues that are required to improve by the authors.

1. The title of this manuscript is related to the cancer image classification, but the datasets are only from the breast cancer. The author should point out the specific cancer type throughout the context.

2. In this paper, the author report two novelty, but in the abstract they only present the second one, and just simply mentioned the HE image normalization. they may also give more information for the first one, which may be interested by many readers.

3. The introduction has the limited background, and need to add more literature discussion and introduce current issues in this field. Especially, any supports related to the improvement of DL normalization?

4. Table 1 is not clear, what are the meanings of the size, parameters, and depth? 

5. Most parts of section 3.1 should be moved to the Methods and Materials, not belonged to the results.

6. The detailed training parameters and hardware environment are missing, for example the learning rate, pitch size, and so on. Because they are very important in the evaluation of complexity analysis.

Author Response

Reviewer’s Comments 1: The title of this manuscript is related to the cancer image classification, but the datasets are only from breast cancer. The author should point out the specific cancer type throughout the context.

Authors reply:
Thank you for your valuable feedback on the title and dataset used in our manuscript. We truly appreciate your thoughtful consideration of this matter.

You are correct in pointing out that the title of our manuscript pertains to cancer image classification, with a primary focus on breast cancer datasets. We apologize for any confusion this may have caused. We understand the importance of accurately representing the specific cancer type throughout the context. However, we want to clarify that the procedures and methods described in our research are not limited to breast cancer classification alone; they can be applied to other cancer types as well. The reason for selecting breast cancer datasets was the challenge of finding freely available datasets with high stain variation and representation of multiple cancer stages, which is not readily available in many other freely accessible cancer datasets.

In the relevant sections of the paper, we will clearly indicate that we used a breast cancer image dataset while emphasizing the potential applicability of the methodology to other cancer types.

Reviewer’s Comments 2: In this paper, the author reports two novelties, but in the abstract, they only present the second one and just simply mention the HE image normalization. They may also give more information for the first one, which may be interested in by many readers.

Authors reply:
Our primary objective in this research was not to develop a novel GAN model for stain normalization or to optimize existing DL models. We used three stain normalization GANs initially, and based on their performances, we selected the best model (Table 2 provides the GAN models' performances). Instead, our primary focus lies in examining how stain normalization impacts well-known existing DL architectures. Specifically, we are interested in understanding the effects of stain normalization on the performance of less complex, lightweight DL models.

The abstract's emphasis is intended to highlight the broader implications of our research, which aims to shed light on the significant impact of stain normalization on existing DL architectures. By presenting the results and insights related to the performance improvements achieved via stain normalization procedures on lightweight models, we believe we provide valuable information for researchers and practitioners working in cancer image classification.

Reviewer’s Comments 3: Table 1 is not clear; what are the meanings of the size, parameters, and depth?

Authors reply:
Allow us to provide a more detailed explanation of the meanings of the size, parameters, and depth columns in Table 1.

1. Size (MB): The "Size" column represents the model size in megabytes (MB). It refers to the memory footprint of the deep learning model when stored in a file on disk. Model size is an essential factor to consider, especially when deploying models on resource-constrained devices or dealing with large-scale deployments. Smaller model sizes are generally preferred for their reduced memory requirements, enabling more efficient deployment and execution.

2. Parameters: The "Parameters" column indicates the number of trainable parameters in the deep learning model. These parameters are the learnable weights and biases that the model updates during the training process to minimize the loss function and make accurate predictions. The number of parameters is a critical metric in understanding the model's complexity and capacity to learn from the data. Larger models with more parameters tend to have higher learning capacity but may require more data and computational resources for training.

3. Depth: The "Depth" column refers to the depth of the deep learning model, which represents the number of layers in the architecture. Deep learning models with a greater number of layers are often referred to as "deep" models, while those with fewer layers are considered "shallow" models. Model depth is an important aspect of architectural design and influences the model's ability to capture hierarchical features from input data. Deeper models may have better representation learning capabilities but can also be more computationally intensive.

In the revised version of the manuscript, we updated column names in Table 1.

Reviewer’s Comments 4: Most parts of section 3.1 should be moved to the Methods and Materials, not belong to the results.

Authors reply:
We have updated the manuscript accordingly.

Reviewer’s Comments 5: The detailed training parameters and hardware environment are missing, for example, the learning rate, patch size, and so on. Because they are very important in the evaluation of complexity analysis.

Authors reply:
Thank you for your feedback on the reproducibility of our methodology. We have already included comprehensive details in the manuscript, such as hardware specifications, training parameters, and the absence of changes to the original deep learning architectures. If you have any specific questions or concerns, please let us know. Your input is essential to improving the quality of our work.

Round 2

Reviewer 1 Report

Great effort in updating the article. Appreciable

Reviewer 2 Report

I have carefully read the revised version of the manuscript along with the rebuttal letter. I confirm that the authors have made significant effort to address my concerns. I enjoyed reading the detailed clarifications and justifications provided by the authors. I must admit that I am not entirely convinced of the high importance attributed to patch/image size over other several critical factors that are not considered in the implementations. Nevertheless, I respect their logical reasoning and still find their detailed explanations reasonable. Therefore, I believe it would be best to leave the final judgment about this matter to the readers.

Regarding the manuscript itself, as mentioned in my previous round of review, it addresses an interesting problem in the field, and especially after the revision, its quality has improved significantly. I have no further suggestions or remarks for this paper and believe it is worthy of consideration for publication. I once again thank the authors for their thorough explanations and hope that their work will gain attention within the community.

The quality of English is significantly better compared to the previous version of the paper.

Reviewer 3 Report

all of my concerns have been addressed. Thank you.